# Predictive Value of Measures of Vascular Calcification Burden and Progression for Risk of Death in Incident to Dialysis Patients

**DOI:** 10.3390/jcm10030376

**Published:** 2021-01-20

**Authors:** Antonio Bellasi, Luca Di Lullo, Domenico Russo, Roberto Ciarcia, Michele Magnocavallo, Carlo Lavalle, Carlo Ratti, Maria Fusaro, Mario Cozzolino, Biagio Raffaele Di Iorio

**Affiliations:** 1Department of Research, Innovation and Brand Reputation, ASST Papa Giovanni XXIII, 24127 Bergamo, Italy; 2Department of Nephrology and Dialysis, Ospedale Parodi, Delfino, 00034 Colleferro, Italy; dilulloluca69@gmail.com; 3Department of Nephrology, School of Medicine, University “Federico II”, 80125 Napoli, Italy; domenicorusso51@hotmail.com; 4Departments of Vetecerinary Medicine and Animal Productions, University of Naples Federico II, 80137 Naples, Italy; roberto.ciarcia@unina.it; 5Department of Clinical, Internal, Anesthesiology and Cardiovascular Sciences, Policlinico Universitario Umberto I, Sapienza University of Rome, 00185 Roma, Italy; michelefg91@gmail.com (M.M.); carlolavalle@yahoo.it (C.L.); 6Department of Cardiology, Ospedale Ramazzini, 41012 Carpi, Italy; ratticarlo@hotmail.com; 7National Research Council (CNR)–Institute of Clinical Physiology (IFC), 56124 Pisa, Italy; dante.lucia11@gmail.com; 8Department of Medicine, University of Padua, 35122 Padua, Italy; 9Renal Division, Department of Health Sciences, ASST Santi Paolo e Carlo, University of Milan, 20142 Milan, Italy; mario.cozzolino@unimi.it; 10Nefrology and Dialysis, AORN “San Giuseppe Moscati”, 83100 Avellino, Italy; br.diiorio@gmail.com

**Keywords:** vascular calcification, coronary artery calcification, aorta calcification, hemodialysis, risk prediction

## Abstract

Background: Vascular calcification (VC) is a marker of cardiovascular (CV) disease and various methods allow for presence and extension assessment in different arterial districts. Nevertheless, it is currently unclear which one of these methods for VC evaluation best predict outcome and if this piece of information adds to the predictive value of traditional CV risk factors in patients receiving hemodialysis (HD). Methods: data of 184 of the 466 patients followed in the Independent study (NCT00710788) were post hoc examined to assess the association three concurrent measures of vascular calcification and all-cause survival. Specifically, coronary artery calcification (CAC) was determined by the Agatston and the volume score while abdominal aorta calcification was determined by plain X-ray of the lumbar spine (Kauppila score (KS)). Survival and regression models as well as metrics of risk recalculation were used to test the association of VC and outcome beyond the Framingham risk score. Results: Middle-age (62.6(15.8) years) men (51%) and women (49%) starting HD were analyzed. Over 36 (median 36; interquartile range: 8–36) months of follow-up 69 patients expired. Each measure of VC (CAC or KS) predicted all-cause mortality independently factors commonly associated with all-cause survival (*p* < 0.001). Far more importantly, each measurement of VC significantly improved risk prediction and patient reclassification (*p* < 0.001) beyond traditional cardiovascular risk factors. Conclusions: Overall, presence and extension of VC, irrespective of the arterial site, predict risk of all-cause of death in patients starting hemodialysis. Of note, both CAC and KS increase risk stratification beyond traditional CV risk factors. However, future efforts are needed to assess whether a risk-based approach encompassing VC screening to guide HD patient management improves survival.

## 1. Introduction

Vascular calcification (VC) is a useful marker of cardiovascular disease and several methods are available for the assessment of their presence and extension [1,2,3,4,5]. Although the pathogenesis of VC is not well established, several studies suggest that the prevalence of VC increases as renal function declines, likely due to the many metabolic abnormalities that characterize chronic kidney disease (CKD) [3,5,6]. Irrespective of the noxious mechanism responsible for VC deposition and progression, the data suggest that the risk of unfavorable outcome is higher for greater VC burden [2,5]. As alluded by the Kidney Disease Improving Global Outcome (KDIGO) clinical guideline for Chronic Kidney Disease and Mineral Metabolism Disease (CKD-MBD) management, VC, as a marker of vascular disease, may allow for risk prediction refinement as well as for individualized patients management [7].

Although coronary artery calcification (CAC) has been traditionally used to detected presence and extension of vascular calcification, several other less expensive and widely available tools are available to assess VC in different arterial sites as well as vascular risk [2,5]. In these regards, the Kauppila score (KS) using lateral-lateral plain X-ray of the lumbar spine has been proposed to evaluate VC in the abdominal aorta. A few reports suggest a close correlation of KS and CAC as well as KS and risk of death in dialysis patients [1,2,3,4,5]. However, which VC measures best predicts long-term survival and whether a measure of vascular calcification adds to the predictive value of traditional Framingham risk stratification in incident to hemodialysis (HD) patients, has not been determined through a concurrent comparison of these measures in a single prospective cohort [5].

For the present study we examined the association of CAC evaluated by 2 different scores systems namely the Agatston score [8] and the volume score [9], as well as Abdominal Aorta Calcification (AAC) evaluated via lateral-lateral plain X-ray of the lumbar spine (KS) [10] and the risk of all-cause mortality. Metrix of risk prediction reclassification according to presence end extension of VC are also investigated for each of the three measures of VC.

## 2. Material and Methods

### 2.1. Study Cohort and Endpoint of Interest

We utilized data from patients incident to hemodialysis recruited in the INDEPENDENT study (ClinicalTrials.gov: NCT00710788) [11,12]. Briefly, the Independent study was designed to assess the impact of 2 different phosphate binder regimens (calcium free vs. calcium containing-phosphate binder) on Cardiovascular (CV) events as well as all-cause mortality [11,12]. In the Independent study 466 adult patients (>18 years) new to hemodialysis (requiring dialysis < 120 days) were randomized in a 1:1 fashion at 18 dialysis center in Italy to receive open-label sevelamer or calcium carbonate/calcium acetate as phosphate binder [11,12]. Age older than 75 years, history of cardiac arrhythmia, syndrome of congenital prolongation of the QT segment interval, a corrected QT (QTc) longer than 440 ms or increased QT dispersion (QTd), history of coronary artery bypass (CABG), liver dysfunction and hypothyroidism and use of drugs that prolong the QT interval were regarded as exclusion criteria [11,12]. Enrollment began in September 2006 and continued through July 2008 [11,12]. Study follow-up ended in July 2011 [11,12]. Written informed consent was obtained from all participants prior to study entry and after approval from each institutional Ethical Review Board. The study was conducted in adherence to the Declaration of Helsinki, Ethical Principles for Medical Research Involving Human Subjects [11,12].

As part of the study protocol, CAC was evaluated at baseline and after 12 months from study inception [11,12]. Furthermore, in a subgroup (*n* = 184, 39% of the Independent study cohort) of patients AAC was evaluated at study inception according to the method described by Kauppila and coworkers. Of importance, this last evaluation was not mandate by the study protocol and left at the physician discretion.

The endpoint of interest was defined as all-cause of mortality and, by study design, all patients were followed until death or study completion (36 months of follow-up) [11,12].

During follow-up, the investigators were instructed to control blood pressure (blood pressure target: below 130/80 mm Hg), anemia (Hb below 11 g/dL, TSAT below 20%), acidosis (HCO3 between 20 and 24 mmol/L), diabetes (HbA1c < 7.0%), dyslipidemia (total cholesterol below 200 mg/dL; LDL cholesterol below 100 mg/dL; triglycerides below 180 mg/dL), and the parameters of bone mineral metabolism (serum phosphorous 2.5–5.0 mg/dL, serum calcium 8.0–9.9 mg/dL, and intact-PTH between 150–300 pg/mL) according to guidelines available at the time the study was conceived [11,12].

### 2.2. Demographic, Clinical and Laboratory Characteristics as Well as Vascular Calcification Assessment

Demographic, clinical and laboratory characteristics were collected at the Independent study initiation [11,12]. Risk of atherosclerotic event were assessed via the Framingham risk score, as previously described [13]. History of atherosclerotic disease (ASCVD) was defined if any of the following clinical data was reported: history of cerebrovascular disease; peripheral vascular disease; angina pectoris; history of myocardial infarction; aortic aneurysm; history of percutaneous coronary angioplasty with or without stenting.

Routine biochemical laboratory measurements were obtained at baseline and at 6-monthly intervals. For current analysis only data on baseline measurements are considered. All blood samples were taken before the midweek dialysis session and after 12 h fasting. Serum parameters of mineral metabolism, electrolytes, anemia and dialysis adequacy were performed by the usual laboratories of the facilities [11,12].

Vascular calcification and arterial stiffness were evaluated at study entry and at 6-monthly intervals for the first 24 months of study follow-up [11,12]. CAC was assessed by a multislice lightspeed (GE Medical Systems) equipment at one center (Solofra, Italy). A standard imaging protocol was used to acquire a set of ECG-gated tomographic slices from the carina to the diaphragm. For each area of interest identified along the course of the coronary arteries was calculated as originally described by Agatston et al. (Agatston score) [8] as well as Callister et al. (volume score) [9]. To reduce inter-reader variability, all CAC scores were obtained in a single central location. No scan inter- as well as intra-reader variability assessment was performed in the Independent study. However, the reported variability for the Agatston and volume score is about 8–10% [1].

AAC was evaluated by lateral-lateral plain roentgenography via a technique similar to that described by Kauppila et al [10]. In brief, a lateral plain radiograph of the abdomen incorporating the first 4 lumbar vertebrae (L1 to L4) was obtained. The aorta was identified as the tubular structure coursing in front of the anterior surface of the spine and VC scored according to the length of each calcified plaque identified along the anterior and posterior profile of the aorta (points were assigned from 1 to 3: 1; small; 2 moderate; 3: large). With this numerical grading, the score could vary from a minimum of 0 to a maximum of 24 points (Kauppila score (KS)) [11]. Due to the fact that KS was not protocolized, lateral-lateral plain X-rays of the lumbar spine were performed at each participating center and read by the attending physician. Inter- and intra- reader agreement for the KS ranges between 0.71–0.85 and 0.93–0.95, respectively [11].

Arterial stiffness was assessed through carotid-femoral Pulse Wave Velocity (PWV) measurement. PWV was evaluated by applanation tonometry with Pulse Pen (Diatecne, Milan, Italy) as previously described as the ratio between the distance (meter) and the travelling time (second) of the pulse generated each cardiac cycle. Hence, PWV is expressed as m/s and higher value represent stiffer arteries [14].

## 3. Statistical Analysis

In this post-hoc analysis, no adjustment for multiple comparisons was made. Data are expressed as mean (Standard deviation–SD) or median [interquartile range] when appropriate. Categorical variables are presented as number of patients (%). Parametric and non-parametric tests were used to compare demographic and clinical characteristics according to the occurrence of any lethal event before study completion (36 months follow-up). Owing to the skewness of the CAC score, baseline CAC was log transformed [log (CAC +1)] if used as continuous variable or categorized as CAC = 0, CAC between 1–99, CAC between 100–399 or CAC = 400+ if used as categorical variable [15]. The semi-quantitative Kauppila score is represented as continuous variable or quartiles of the study distribution when appropriate. Comparison between patients with and without AAC evaluation are presented in the Appendix A.

To gauge the association between different measures of vascular calcification (i.e., CAC evaluated via Agatston as well as Volume score and AAC) and all-cause mortality, survival analyses were used. Cumulative mortality curves were calculated by VC burden category using the Kaplan–Meier product-limit method while the hazard of all-cause death associated with VC burden as continuous variable was estimated using Cox survival analyses. A multi-step approach was carried out. First, we identified independent predictors of all-cause mortality in the study cohort. All variables associated with death at univariate analyses (variable with *p* < 0.15) and based on available evidence were forced into a multivariable adjusted Cox-model (Appendix A). Then, a stepwise approach was employed to identify the most parsimonious model to predict the outcome of interest (Appendix A). Subsequently, the independent association of each individual measurement of VC and all-cause mortality was tested against the independent predictors of death (fully adjusted model: adjusted for measure of VC, Pulse Wave Velocity, age, Framingham score, diabetes, ASCVD, systolic blood pressure, serum levels of phosphate, calcium, PTH, use of ARBs, beta-blockers, vitamin D, calcium containing phosphate binder, calcium channel blockers and cinacalcet). Consistently with the methodology used to identify predictors of all-cause mortality, the most parsimonious model to predict death was identified through a stepwise procedure starting from the fully adjusted model.

Logistic regression was utilized to evaluate the incremental ability of each measurement of VC to predict all-cause mortality by computing the C-index, calibration statistic, integrated discrimination improvement (IDI) and continuous net reclassification improvement (NRI) of the most parsimonious model reported in with or without measurement of VC. Further, the discriminative value of the model with and without the measurement of VC was also investigated by means of receiver operating characteristic (ROC).

Statistical significance was set at 0.05. All analyses were completed using R version 3.6.2 (21 December 2019; The R Foundation for Statistical Computing, Vienna, Austria).

## 4. Results

Overall, we investigated 184 middle age subjects (39% of the study cohort of the Independent study) with measurement of CAC and AAC at study inception available. The main characteristics of the study cohort are reported in Table 1. Because AAC assessment was not mandatory and left at the physician discretion some differences between subject recruited in the Independent study with and without AAC assessment are present (Appendix A).

Over a median follow-up of 36 [interquartile range IQR: 8–36] months, 69 patients expired. Patients who died of all-cause during follow-up were older [69(14) vs. 58(15)], exhibited higher systolic blood pressure [139(17) mmHg vs. 132(18) mmHg], Framingham risk score [13.0(2.9) vs. 11.1(3.7)], coronary as well as aortic aorta calcium burden and arterial stiffness (Table 1). Furthermore, differences in CKD-MBD as well as treatments were also apparent (Table 1).

To gauge the independent contribution of all factors associated with mortality, all variables significantly associated with death at univariate analyses (Table 1A, variable with *p* < 0.15) were forced into a multivariable adjusted Cox-model (Appendix A) and the most parsimonious model was selected via a stepwise method (Appendix A). Accordingly, increased CAC burden and pulse wave velocity, history of diabetes mellitus and use of calcium containing phosphate binders were among the others the most relevant factors associated with risk of death (Appendix A).

Notably, at unadjusted analysis all 3 measurements (CAC assessed via the Agatston and volume score as well as AAC) were significantly associated (*p* < 0.001) with the risk of all-cause mortality Figure 1A–C.

Nonetheless, the independent association of CAC or KS was tested against the predictors of death Table 2A–C. In general, measures of CAC either assessed via the Agatston or the volume score systems and AAC were independently and strongly related with risk of death irrespectively of the baseline atherosclerotic risk assessed via the Framingham score or history of ASCVD (Table 2A–C). Furthermore, addition of VC increased the overall performance of each model (significant AIC increase with addition of measurement of VC to the model) Table 2A–C.

Furthermore, logistic regression models encompassing information regarding VC either in the coronary or aortic arteries allowed for a significant improvement in risk prediction (Table 3A–C). Indeed, addition of measurement of VC to the model, increased discrimination (C-statistics) and allowed a significant risk reclassification (IDI and NRI) of subjects new to hemodialysis (*p* < 0.001) (Table 3A–C). However, in this study cohort the logistic model that included the Agatston CAC score was not well calibrated (Calibration Chi-square; *p* value < 0.001) and results may be less accurate that for other measurements of VC (Table 3A–C and Figure 2A–C).

## 5. Discussion

VC is a complex phenomenon involving active and passive mechanisms such as dysregulation of mineral metabolism [3,5,7,16,17,18] As renal function declines, prevalence of VC increases ranging from 40% in CKD stage 3 and 4, to 60% in CKD stage 5 patients entering dialysis and 80–85% in prevalent hemodialysis patients [5,7]. Furthermore, data suggest a link of VC with poor survival and a variety of cardiovascular events such as acute and chronic coronary artery syndromes, heart failure, cardiac arrhythmias and sudden death [5,7,18]. Because CKD is also associated with excessive vascular risk, it is logical to assume that VC assessment may improve risk stratification and possibly personalized treatment(s) in subjects with impaired renal function. However, only few studies have investigated the predictive value of VC screening in different arterial sites (i.e., coronary arteries and abdominal aorta) in a cohort of patients starting dialysis.

Overall, the main findings of this study are that presence and extension of VC predicts risk of death in patients starting hemodialysis irrespective of the arterial site. Even though CAC quantitatively measures VC and is commonly perceived as a better predictor of outcome than abdominal aorta calcification, the difference is minimal and all measurements of VC allowed for a significant risk re-stratification beyond traditional risk factors for CV disease (i.e., Framingham risk score) or history of atherosclerotic vascular disease.

Current findings expand available evidence. Previous experience is limited to one method for VC assessment or have not evaluated the incremental predictive value of VC beyond traditional CV risk factors [5,18]. Furthermore, current results support the use of CAC or KS to identify high risk and fragile patients as suggested by the KDIGO clinical guidelines on CKD-MBD [7]. Indeed, a risk-based approach encompassing markers of vascular disease to guide management in patients receiving dialysis is advisable in light of the high vascular risk and poor predictive value of traditional CV risk factors in these subjects [17]. In these regards, CAC or KS are simple tools to assess the individual vascular risk.

Irrespective of the metabolic pathway or mechanism promoting VC, CAC and AAC are markers of tissue damage and their presence suggest a prolonged exposure to a vascular *noxa* that ultimately results in deposition of crystals of hydroxyapatite in the context of the arterial wall [3,5,18]. While circulating biomarkers reflect the instantaneous risk, tissue markers reflect the cumulative exposure to noxious factors to which an individual has been exposed [3]. In these regards, VC are less susceptible to transient perturbation of the subject’s homeostasis and are stronger predictors of risk than circulating biomarkers.

Numerous techniques are available to measure VC [1,5]. Simple in office tools such as plain X-rays, 2D ultrasound and echocardiography or more sophisticated and such as electron beam computed tomography and multi-slice computed tomography can be used to detect and quantify VC in blood vessels as well as cardiac valves [1,5]. Quantification of VC with computed tomography (CT) is currently the gold standard to score VC. In this study, we compared the predictive value of two different CT scoring systems [8,9] for evaluating CAC and AAC assessed according to the methodology described by Kauppila et al [10]. While both the Agatston and the volume score quantitatively assess CAC [8,9], the KS semi-quantitatively (score range 0–24) assess the abdominal aorta coursing in front of the lumbar spine [10]. However, CT exposes the patients to a greater radiation burden and is less available than plain X-ray used for KS [1].

Although the peculiar reasons for arterial vulnerability in CKD patients is not completely understood, several lines of evidence confirmed the association of CAC and KS and risk of death or CV events in different stages of CKD including patients on maintenance dialysis [5]. Overall, a graded increase in the risk of unfavorable prognosis is associated with progressively higher burden of calcification, irrespective of adjustments for confounders [5,15]. However, while previous observations have suggested a strong accordance among these measurements [1], no study has investigated whether these tools have different predictive value and which one should be used for risk prediction. Current results, suggest that independently of the arterial site investigated or the scoring system utilized, models encompassing VC significantly (*p* < 0.001) improve risk reclassification supporting the use in clinical practice of simple in-office tools such as plain X-ray for vascular calcification detection.

This piece of information may be of particular use in consideration of the potential therapeutic implication. Indeed, some lines of evidence suggest that CKD-MBD or anticoagulant management may impact the risk of VC deposition and progression as well as the risk of unfavorable outcomes [7,19,20,21]. Further, a recently published study showed that a newer compound SNF472 significantly attenuates CAC progression in hemodialysis patients [22], independently of baseline characteristics [23]. However, whether VC attenuation improves survival or reduces CV events await definitive confirmation in future studies.

This study has several limitations. The relatively small sample size and the retrospective nature are among the others the most relevant ones. Indeed, KS was not protocolized nor centralized and some differences between patients with and without this assessment in the Independent study cohort exist (Appendix A). While these limit the generalizability of current findings, a rigorous method was used to control for potential confounders. Furthermore, the fact that all three measurements of VC yielded similar results suggest that the reported findings are consistent.

In conclusion, we documented in a cohort of patients new to dialysis that VC predict outcome and significantly improve risk stratification beyond traditional CV risk factors. Overall, it seems that CAC is a better predictor of outcome than abdominal aorta VC, though the difference is minimal. However, future efforts are needed to assess whether a risk-based approach encompassing VC screening to guide HD patient management improves survival.

## Figures and Tables

**Figure 1 jcm-10-00376-f001:**
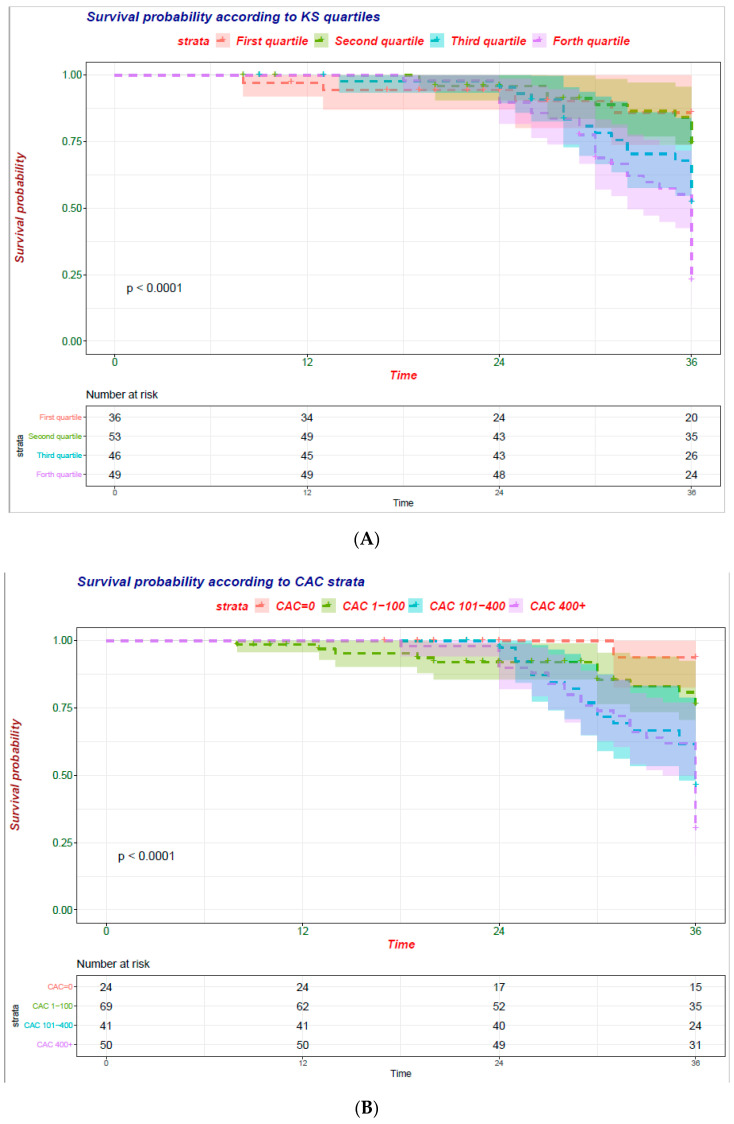
Cumulative mortality curves were calculated by VC burden category according to the Kaplan–Meier product-limit method. (**A**) Abdominal Aorta Calcification (AAC) evaluated by quartiles of Kauppila score (KS); (**B**) Coronary Artery Calcification (CAC) evaluated via the Agatston score; (**C**) CAC evaluated via the volume score.

**Figure 2 jcm-10-00376-f002:**
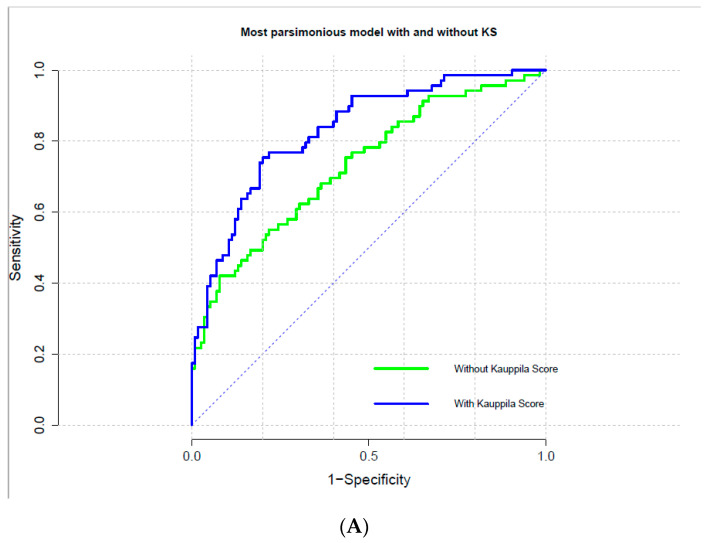
Receiver operating characteristic curves of most parsimonious models to predict all-cause mortality with and without measurement of VC. (**A**) Abdominal Aorta Calcification (AAC) evaluated by quartiles of Kauppila score (KS); (**B**) Coronary Artery Calcification (CAC) evaluated via the Agatston score; (**C**) CAC evaluated via the volume score.

**Table 1 jcm-10-00376-t001:** Patients Characteristics according to study cohort and status at study completion.

	Total (*n* = 184)	Alive (*n* = 115)	Expired (*n* = 69)	
Variable	Mean (SD) or *n* (%)	Mean (SD) or *n* (%)	Mean (SD) or *n* (%)	*p*-Value
**Demographic and Clinical Characteristics**				
Age (years)	62.6 (15.8)	58.5 (15.1)	69.4 (14.5)	<0.0001
Male (%)	94(51.0%)	57 (49.5%)	37(53.6%)	0.703
Body Weight (Kg)	68.3 (13.1)	69.2 (13.5)	66.7 (12.4)	0.198
ASCVD (%)	27(14.6%)	14 (12.1%)	13(18.8%)	0.307
Diabetes (%)	40 (21.7%)	22(19.1%)	18(26.0%)	0.356
Systolic Blood Pressure (mmHg)	135 (18)	132 (18)	139 (17)	0.012
Diastolic Blood Pressure (mmHg)	75 (9)	75 (9)	75 (10)	0.978
Framingham score (unit)	11.8 (3.5)	11.1 (3.7)	13.0 (2.9)	<0.001
**Measurement of Vascular Calcification & Arterial Stiffness**				
CAC Agatston score	569 (1098)	226 (579)	1139 (1468)	<0.0001
CAC Agatston score progression	-	-	-	
CAC Volume score	229 (334)	112 (223)	423 (393)	<0.0001
Abdominal Aorta VC (Kauppila score)	13 (9)	10 (8)	18 (7)	<0.0001
Pulse Wave Velocity (m/s)	9.5 (3.7)	9.2 (3.7)	10.0 (3.8)	0.156
**Laboratory Characteristics**				
Albumin (g/dL)	3.7 (0.4)	3.7 (0.3)	3.7 (0.4)	0.794
Creatinine (g/dL)	7.9 (2.5)	8.0 (2.4)	7.6 (2.8)	0.376
Hemoglobin (g/dL)	11.0 (1.6)	11.1 (1.7)	10.9 (1.3)	0.428
Sodium (mE/L)	139 (3.5)	139 (3.7)	139 (2.9)	0.152
Potassium (mEq/L)	5.1 (0.7)	5.0 (0.7)	5.2 (0.7)	0.175
Calcium (mg/dL)	8.8 (0.9)	8.9 (0.9)	8.6 (0.7)	0.036
Phosphate (mg/dL)	4.5 (1.3)	4.4 (1.2)	4.8 (1.4)	0.055
Parathyroid Hormone (pg/mL)	259 (227)	236 (180)	298 (287)	0.111
C-reactive protein (mg/L)	5.0 (3.3)	4.9 (3.6)	5.1 (2.8)	0.762
**Concomitant Medications**				
Use of ACE-inhibitors (%)	132 (71.7%)	78(67.8%)	54(78.2%)	0.176
Use of ARBs (%)	148(80.4%)	87(75.6%)	61(88.4%)	0.055
Use of betablockers (%)	115 (62.5%)	81(70.4%)	34 (49.2%)	0.007
Use of calcium channel blockers (%)	56 (30.4%)	27 (23.4%)	29 (42.0%)	0.013
Use of cinacalcet (%)	79 (42.9%)	44 (38.2%)	35 (50.7%)	0.134
Use of vitamin D (%)	111 (60.3%)	78 (67.8%)	33(47.8%)	0.011
Use of Sevelamer (%)	29(15.7%)	23(20.0%)	6 (8.7%)	0.067
Use of calcium based binders (%)	155 (84.2%)	92(80.0%)	63 (91.3)	0.067

ASCVD: atherosclerotic cardiovascular disease defined if any of the following clinical data was reported: history of cerebrovascular disease; peripheral vascular disease; angina pectoris; history of myocardial infarction; aortic aneurysm; history of percutaneous coronary angioplasty with or without stenting; CAC: coronary artery calcification; VC: vascular calcification; ARB: angiotensin receptor blocker.

**Table 2 jcm-10-00376-t002:** Independent predictors of all-cause mortality in the study cohort 1 according to measurement of Vascular calcification. (**A**) Abdominal Aorta Calcification (AAC) evaluated via the Kauppila score; (**B**) Coronary Artery Calcification (CAC) evaluated via the Agatston score; (**C**) CAC evaluated via the volume score. For each measurement of VC, the most parsimonious survival Cox models selected via a stepwise approach is presented. The fully adjusted model included predictors of all-cause mortality identified (reported in Appendix A).

**(A) Abdominal Aorta Calcification (AAC) Evaluated via the Kauppila Score**				
**Variable**	**HR**	**Lower 0.95**	**Upper 0.95**	**Pr(>|z|)**
Kauppila score (1U increase)	1.095	1.0577	1.133	<0.001
Pulse wave velocity (m/s)	1.061	0.9961	1.13	0.0658
Age (years)	1.019	1.0002	1.038	0.0473
Systolic blood pressure (mmHg)	1.013	0.9987	1.027	0.0747
Use of calcium channel blockers (y vs. n)	1.476	0.8923	2.44	0.1295
**Model fit Statistics (AIC-adding Kauppila)**	624.31 (final model with Kauppila)			
loglink (without Kauppila)	332.34			
logling (with Kauppila)	316.34			
Comparison with vs. without Kauppila score	Chisq 31.89 (*p* < 0.001)			
**(B) Coronary Artery Calcification (CAC) Evaluated via the Agatston Score**				
**Variable**	**HR**	**Lower 0.95**	**Upper 0.95**	**Pr(>|z|)**
CAC-Agatstone score (log increase)	1.6279	1.4176	1.869	<0.001
Pulse wave velocity (m/s)	1.1023	1.011	1.202	0.0273
Diabetes (y vs. n)	3.597	1.7437	7.42	0.00053
ASCVD (y vs. n)	0.5582	0.2718	1.146	0.11226
Systolic blood pressure (mmHg)	1.011	0.9974	1.025	0.11198
Use of calcium containing phosphate binder (y vs. n)	2.9523	0.9032	9.65	0.07321
Use of calcium channel blockers (y vs. n)	1.9427	1.1263	3.351	0.01696
**Model fit statistics (AIC-adding CAC)**	624.31 (final model with CAC)			
loglink (without CAC)	305.16			
logling (with CAC)	333.41			
Comparison with vs. without CAC score	Chisq 43.697 (*p* < 0.001)			
**(C) Coronary Artery Calcification (CAC) Evaluated via the Volume Score**				
**Variable**	**HR**	**Lower 0.95**	**Upper 0.95**	**Pr(>|z|)**
CAC-Volume score (log increase)	1.7301	1.4469	2.069	<0.001
Pulse wave velocity (m/s)	1.0968	1.0082	1.193	0.03158
Age (years)	1.0167	0.9967	1.037	0.10152
Diabetes (y vs. n)	3.1042	1.4553	6.622	0.00338
ASCVD (y vs. n)	0.5692	0.282	1.149	0.11584
Systolic blood pressure (mmHg)	1.0103	0.9966	1.024	0.13968
Use of calcium containing phosphate binder (y vs. n)	2.6029	0.8045	8.421	0.11029
Use of calcium channel blockers (y vs. n)	1.6822	0.9516	2.974	0.07356
**Model fit statistics (AIC-adding CAC)**	627.21 (final model with CAC)			
loglink (without CAC)	305.6			
logling (with CAC)	327.45			
Comparison with vs. without CAC score	Chisq 43.697 (*p* < 0.001)			

HR: Hazard Ratio; lower 0.95: lower boundary of the 95% Confidence Interval; upper 0.95 upper boundary of the 95% Confidence Interval; ASCVD atherosclerotic cardiovascular disease defined if any of the following clinical data was reported: history of cerebrovascular disease; peripheral vascular disease; angina pectoris; history of myocardial infarction; aortic aneurysm; history of percutaneous coronary angioplasty with or without stenting; AIC: Akaike information criterion.

**Table 3 jcm-10-00376-t003:** Logistic regression was utilized to evaluate the incremental ability of each measurement of VC to predict all-cause mortality by computing the C-index, calibration statistic, integrated discrimination improvement (IDI) and continuous net reclassification improvement (NRI) of the most parsimonious model reported in Table 2A–C with or without measurement of VC. (**A**) Abdominal Aorta Calcification (AAC) evaluated via the Kauppila score; (**B**) Coronary Artery Calcification (CAC) evaluated via the Agatston score; (**C**) Coronary Artery Calcification (CAC) evaluated via the volume score.

**(A) Abdominal Aorta Calcification (AAC) Evaluated via the Kauppila Score**			
**Best Cutoff to Discriminate Expired vs. Alive Patients at Univariate Analyses**			
14.5 (specificity 67.0%–sensitivity 78.3%)			
**Metrics of discrimination-C-statistics (95%CI)**			
without Kauppila	0.730 (0.655–0.806)		
with Kauppila	0.841 (0.782–0.900)		
Difference in C statistics (SD)	0.110 (0.017); *p*-value for comparison < 0.001		
**Model fit statistics (AIC)**			
without Kauppila	222.78		
with Kauppila	186.81		
Comparison between models (ANOVA)	LR (Chisq) 37.9; d.f. 1 (*p* < 0.001)		
**Metrics of calibration (adding Kauppila)**	Chi-square	Df	*p*-value
Hosmer-Lemeshow goodness of fit	8.032	8	0.43
**Patient reclassification (adding Kauppila)**	Coeff	95%CI	*p*-value
IDI (95%CI)	0.177	(0.123–0.232)	*p* < 0.001
NRI categorical (95%CI)	0.246	(0.130–0.361)	*p* < 0.001
NRI continuous (95%CI)	0.921	(0.623–1.220)	*p* < 0.001
**(B) Coronary Artery Calcification (CAC) Evaluated via the Agatston Score**			
**Best cutoff to Discriminate Expired vs. Alive Patients at Univariate Analyses**			
257.5 (specificity 80.9%–sensitivity 73.9%)			
**Metrics of discrimination-C-statistics (95%CI)**			
without CAC	0.742 (0.669–0.8157)		
with CAC	0.901 (0.854–0.947)		
Difference in C statistics (SD)	0.158 (0.026); *p*-value for comparison *p* < 0.001		
**Model fit statistics (AIC)**			
without CAC	225.53		
with CAC	160.18		
Comparison between models (ANOVA)	LR (Chisq) 67.3; d.f. 1 (*p* < 0.001)		
**Metrics of calibration (adding CAC)**	Chi-square	Df	*p*-value
Hosmer-Lemeshow goodness of fit	21.116	8	0.006
**Patient reclassification (adding CAC)**	Coeff	95%CI	*p*-value
IDI (95%CI)	0.311	(0.241–0.381)	*p* < 0.001
NRI categorical (95%CI)	0.301	(0.173–0.430)	*p* < 0.001
NRI continuous (95%CI)	1.275	(0.976–1.573)	*p* < 0.001
**(C) Coronary Artery Calcification (CAC) Evaluated via the Volume Score**			
**Best Cutoff to Discriminate Expired vs. Alive Patients at Univariate Analyses**			
66.5 (specificity 70.4%–sensitivity 81.2%)			
**Metrics of discrimination-C-statistics (95%CI)**			
without CAC	0.776 (0.707–0.845)		
with CAC	0.896 (0.848–0.943)		
Difference in C statistics (SD)	0.120 (0.021); *p*-value for comparison < 0.001		
**Model fit statistics (AIC)**			
without CAC		214.03	
with CAC		162.83	
Comparison between models (ANOVA)	LR (Chisq) 53.1; d.f. 1 (*p* < 0.001)		
**Metrics of Calibration (adding CAC)**	Chi-square	Df	*p*-value
Hosmer-Lemeshow goodness of fit	11.897	8	0.1558
**Patient reclassification (adding CAC)**	Coeff	95%CI	*p*-value
IDI (95%CI)	0.241	(0.177–0.305)	*p* < 0.001
NRI categorical (95%CI)	0.301	(0.173–0.429)	*p* < 0.001
NRI continuous (95%CI)	1.072	(0.774–1.370)	*p* < 0.001

ASCVD: atherosclerotic cardiovascular disease defined if any of the following clinical data was reported: history of cerebrovascular disease; peripheral vascular disease; angina pectoris; history of myocardial infarction; aortic aneurysm; history of percutaneous coronary angioplasty with or without stenting; AIC: Akaike information criterion; IDI: integrated discrimination improvement; NRI: net reclassification improvement.

## Data Availability

The data of the Independent Study are not publicly available due to Data Protection restrictions.

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
