# Peer review of "Predictive Value of Measures of Vascular Calcification Burden and Progression for Risk of Death in Incident to Dialysis Patients"

_jcm, 2021, doi:10.3390/jcm10030376_

Round 1

Reviewer 1 Report

Overall this paper presents a study of important cardiovascular prediction factors and displays a thorough statistical analysis. The main improvements to the paper include rearranging some sections and making text more clear for the reader.

Comments on what to improve:

  1. In the introduction, a few more lines should be dedicated to the abdominal aorta calcification metric. Since this metric is part of the major conclusions of the study, it should be more than briefly explained.
  2. When reading the abstract, it appears there are 184 subjects, but then reading the methods it notes that this is only a subpopulation of the study. This should be further clarified so as not to confuse a reader.
  3. The 'Statistical Analysis' section of the methods is very thorough; some should be moved to an appendix and some graphs belong in the results. The analysis is well done, but goes beyond what fits well in the standard methods section. Instead of the 'Results' section referring back to previously displayed data, these data could be shown for the first time here.
  4. In abstract line 62, it seems like the main finding is that CAC is a better predictor than abdominal aorta VC. However, this is dismissed as not the most significant finding in the discussion section (lines 262-265). I recommend rephrasing the abstract to more clearly lay out the take-home messages for the readers.

Reviewer 2 Report

Antonio et al. provided important clinical findings and presented the results appropriately. One major concern is raised about the interobserver variability. Please provide how the authors manipulated the interobserver variability to calculate the vascular calcification score. How many radiologists were involved in assessing the calcification score? Can you please provide Cohen's k value? This point is one of the most important parts in the study of the calcification score.

Round 2

Reviewer 2 Report

The issue raised by a reviewer was well resolved through the revision.

Author Response

We thank the reviewer for the useful comments.